# Group size and composition influence collective movement in a highly social terrestrial bird

Danai Papageorgiou[1,2,3,4]\*, Damien Roger Farine[1,2,3,5]†

[1]Max Planck Institute of Animal Behavior, Department of Collective Behavior, Universitätsstraße, Konstanz, Germany; [2]University of Konstanz, Department of Biology, Universitätsstraße, Konstanz, Germany; [3]University of Konstanz, Center for the Advanced Study of Collective Behaviour, Universitätsstraße, Konstanz, Germany; [4]Kenya Wildlife Service, Nairobi, Kenya; [5]Department of Ornithology, National Museums of Kenya, Nairobi, Kenya

**Abstract** A challenge of group-living is to maintain cohesion while navigating through heterogeneous landscapes. Larger groups benefit from information pooling, translating to greater 'collective intelligence', but face increased coordination challenges. If these facets interact, we should observe a non-linear relationship between group size and collective movement. We deployed high-resolution GPS tags to vulturine guineafowl from 21 distinct social groups and used continuous-time movement models to characterize group movements across five seasons. Our data revealed a quadratic relationship between group size and movement characteristics, with intermediate-sized groups exhibiting the largest home-range size and greater variation in space use. Intermediate-sized groups also had higher reproductive success, but having more young in the group reduced home-range size. Our study suggests the presence of an optimal group size, and composition, for collective movement.

**\*For correspondence:**
dpapageorgiou@ab.mpg.de

**Present address:** †Department of Evolutionary Biology and Environmental Studies, University of Zurich, Zurich, Switzerland

**Competing interests:** The authors declare that no competing interests exist.

## Introduction

Movement behaviour fundamentally influences how individuals encounter resources that are critical for their fitness. In highly social animals, movements are determined by how efficiently a group can coordinate their behaviour (*Chapman and Chapman, 2000*; *Couzin et al., 2002*; *Strandburg-Peshkin et al., 2017*; *Strandburg-Peshkin et al., 2015*). Coordination requires effectively deciding where and when to move (*Conradt and Roper, 2005*) as well as maintaining cohesion during movement (*Gall and Manser, 2017*). Both these factors could be shaped by intragroup characteristics. For example, it has become increasingly evident that group composition (*Farine et al., 2015*), such as the presence of uniformed individuals (*Couzin et al., 2011*) or the mixture of individual personality types (*Aplin et al., 2014*; *Jolles et al., 2017*), can shape the behavioural properties of groups. In societies where individuals form stable groups, group size is another potentially important group characteristic that may also significantly impact fine-scale collective movement patterns. However, while the balance of benefits (such as decreased predation risk according to the 'many eyes hypothesis'; *Pulliam, 1973*) and costs (increased intra-group competition over resources) associated with group size have been extensively explored (*Krause and Ruxton, 2002*), we still have very little information on whether group size translates to differences in movement behaviour and ranging patterns of naturally occurring groups.

Theory predicts that larger group size can benefit animal groups that move collectively through complex and heterogeneous landscapes. If group members pool their information, then larger groups should be able to reach more accurate estimates of their environment (*Cantor et al., 2020*;

**eLife digest** Many social animals live in stable groups that stay together for years, or even lifetimes. Being in a group offers a range of benefits, such as safety from predators, information on where to find food or water, and more accurate navigation. But these benefits come at a cost. The larger the group, the harder it is to make decisions that balance the needs of each individual. So, while members of a large group should be better at locating resources and finding their way, they may take longer to decide where to go next.

In nature, groups of the same species can vary greatly in size and can have large or small numbers of offspring. This raises the question of whether there is an optimal group size where the benefits of living together are maximized relative to the costs? To help answer this question, Papageorgiou and Farine studied the group behaviour of vulturine guineafowl, a social, ground-dwelling bird found in the savannahs of East Africa.

A lightweight GPS tracker was fitted to the members of 21 different groups of vulturine guineafowl to see how group size affects the movement of these birds. The tags collected data every five minutes from dawn until dusk each day, and remained active over five two-month spans of similar weather conditions. This revealed that groups of intermediate size, which contain 33 to 37 birds, ranged over larger areas allowing them to access more diverse resources, and used less energy by travelling shorter distances. Birds in these groups also explored more new areas, decreasing their chances of encountering a predator, and produced more chicks, suggesting that their collective behaviour gave them a reproductive advantage.

These findings suggest that intermediate sized groups display an optimal level of movement compared to larger or smaller groups. Understanding how social groups of different sizes interact with their environment can aid conservation planning. Future work should focus on how this relationship changes with the seasons. This could reveal more about the effects of group size during challenging conditions, like drought.

Couzin et al., 2011), as highlighted by the 'wisdom of the crowd' theory (Galton, 1907). This process can translate to navigational accuracy, via the 'many wrongs hypothesis', as pooling the imperfect individual estimations of directions can generate accurate group-level estimates that allow groups to move more directly towards resources (Simons, 2004). Larger groups are also more likely to contain individuals with knowledge relevant to the current position or situation of a group (the 'pool of competence hypothesis'; Giraldeau, 1984). For example, studies on killer whales (Brent et al., 2015) and elephants (McComb et al., 2001) have shown that older individuals become leaders under conditions of extreme resource shortages. Such mechanisms should therefore allow larger groups to more effectively navigate their environment, and therefore exhibit a larger home-range size.

Effective navigation, and subsequently larger home ranges, should be beneficial for group members. Larger areas encompass a greater diversity and abundance of resources needed to meet group members' nutritional needs (Corriale et al., 2013; Hubbs and Boonstra, 1998; Maher and Burger, 2011), and moving to more new areas increases the probability of discovering new food resources. Using larger areas can also allow groups to be more variable in their movements, which could make them more unpredictable to predators (Roth and Vetter, 2008) and thus decreases predation risk (Richardson et al., 2018). Thus, all else being equal, such as environmental characteristics, the members of a group with a relatively larger home range should have higher survival in periods of resource limitations and/or greater reproductive output than an identical group with a smaller home range. In reality, home-range size is often limited by many factors, including competition for space with other groups (Christensen and Radford, 2018). However, many group-living species live in non-territorial societies, where groups can benefit from exploiting a home range without paying the costs of inter-group fights and competition. In such species, groups can interact affiliatively and preferentially with specific other groups, resulting in a multilevel society. In such societies, having a larger home-range size might generate more chances to associate with conspecifics from other groups (Grueter et al., 2020), which, in turn, can increase mating opportunities (by increasing contact with members of the opposite sex; Grueter et al., 2015), dispersal opportunities (as sub-adults often disperse to

neighbouring groups; *Städele et al., 2015*) and information sharing about food resources or predation risk (*Whitehead et al., 2012*). Thus, there are many ways that groups can benefit from maximizing their space use, up to the point where the costs of travel and navigation do not override the benefits gained.

However, living in large groups can also introduce challenges, particularly in species that maintain stable group membership. Collective movement requires maintaining cohesion, a challenge that increases with group size and in more complex environments (*Codling et al., 2007*). From a movement perspective, coordination among many individuals could result in slower decision-making, due to both coordination challenges as well as larger groups being likely to harbour more conflicts of interest. For example, one study of collective movement in baboons found that the probability that an individual baboon followed one initiator versus the other was much lower when both disagreement among initiators on where to go and the number of initiators increased (*Strandburg-Peshkin et al., 2015*). Larger groups are also likely to have slower movement speeds while 'on-the-go'. We all have personal experiences of how much slower it can be to move or make decisions (such as where to have lunch) when we are in a large group, as the group generally may have to conform to the slowest members (*Herbert-Read et al., 2013*) or satisfy a diversity of preferences (*Strandburg-peshkin et al., 2017*). Finally, it may be more challenging for individuals to influence larger groups to move to new areas, and thus to extend the group's home-range. These factors suggest that the consensus costs borne by larger groups will ultimately limit the size of the home-range that such groups can exploit.

Given that both positive and negative relationships between group size and home-range size may exist, we a priori predict a non-linear relationship between the size of stable groups and collective movement characteristics. This non-linear relationship is likely to represent an optimal group size for movement that reflects the point where the difference between navigational accuracy and coordination problems is maximized. To our knowledge, this relationship remains largely unexplored in wild, free-living groups. A first challenge in quantifying the relationship between home-range size and group size requires long-term tracking across multiple replicated groups and, ideally, replication in time. A second challenge is that many species that live in stable groups (at least most of the well-studied species) either experience increased intra-group competition as group sizes increase (e.g. baboons; *Markham et al., 2015*) or they are territorial (e.g. meerkats; *Bateman et al., 2015*). Territoriality introduces several confounding effects. For example, the greater resource-holding potential of larger groups (*Bateman et al., 2015*; *Markham et al., 2015*) can translate to larger home-ranges. Further, different territorial groups can vary in the density and distribution of resources within their territory, which could make it appear that smaller groups have larger home-ranges if they live in marginal areas with low resource densities. As a result, the relationship between group movement characteristics and group size has remained relatively unclear.

In this study, we examine the effect of group size on the movement and space use patterns of wild groups of vulturine guineafowl (*Acryllium vulturinum*), and whether resulting patterns correspond to differences in reproductive output among groups. Our study species lives in a multilevel society consisting of large, stable groups with multiple breeding units and groups form stable associations with specific other groups (*Papageorgiou et al., 2019*). The stable social structure and the lack of territoriality—the entire population inhabits a small area of approximately 500 ha and all groups can easily access all parts of our study area—makes vulturine guineafowl an ideal species to investigate how group size influences movement characteristics because groups can range without major constraints on ranging imposed by other groups. Further, their large body size (~1.5 kg) allow guineafowl to be fitted with high-resolution solar-powered GPS tags to collect both fine-scale data on their movement as well as long-term data on their ranging behaviour.

We use data from GPS tags fitted to one individual from a total of 21 distinct groups, collected over five 2-month-long non-breeding seasons (*Figure 1*). We fit a continuous-time movement model (ctmm) to each individual GPS dataset (one bird per group per season) using GPS locations collected every 5 min. From each model, we calculated (1) home-range size, (2) the tendency for groups to use the same areas on consecutive days, and (3) daily travel distance. We then used high-resolution (1 Hz) GPS location data from each individual to quantify (4) the movement speed of its group while 'on the go'. Because groups move through heterogeneous environments, larger groups should be able to navigate more effectively. Their large size should also result in larger daily travel distances as they search for the resources needed to survive. By contrast, moving as a large group also

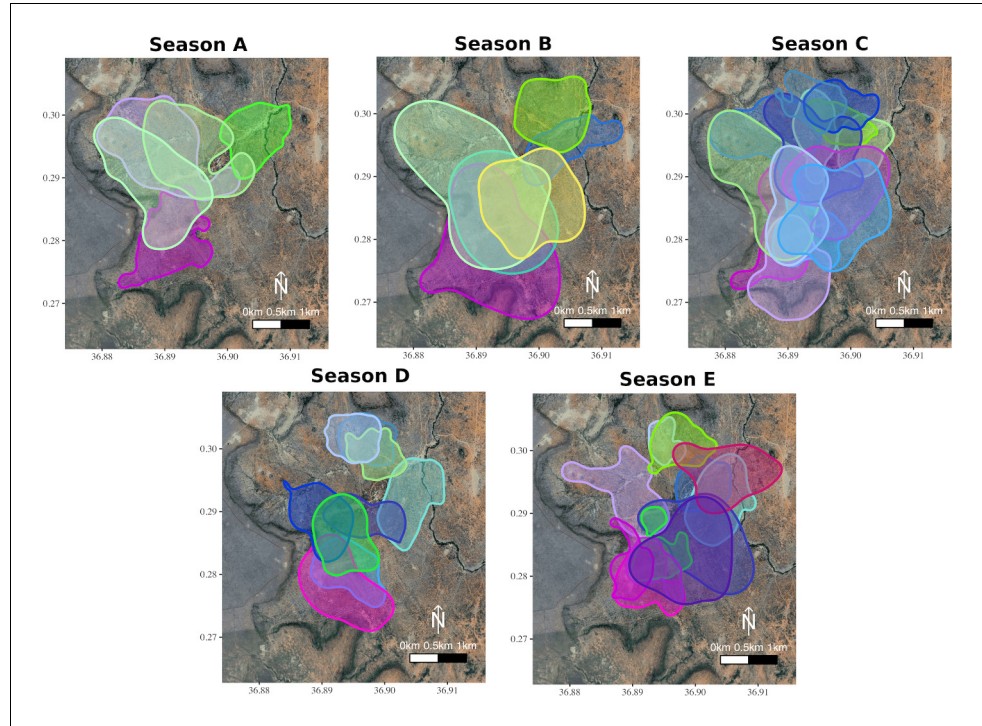

**Figure 1.** Group home-ranges in each of the study seasons. The overlapping home-ranges (95% Auto-correlated Kernel Density Estimate; *Fleming et al., 2015* of each the groups (up to 21 per season) across five different seasons calculated from the ctmms). Seasonal data were collected during periods of two months defined as being intermediate in terms of rainfall and greenness, as groups split to reproduce during twice-yearly breeding seasons and make exceptional movements to find water during droughts (see Materials and methods). These seasons were (A) 30.5.2017 to 1.8.2017, (B) 15.12.2017 to 15.2.2018, (C) 1.8.2018 to 29.9.2018, (D) 10.6.2019 to 9.8.2019, and (E) 14.1.2020 to 16.3.2020. Group membership was determined from daily census observations, and one GPS-tagged individual was selected at random from each group to define the group's movement. Each distinct group is represented by a specific colour, and group colours are kept consistent across seasons.

introduces consensus costs, and these could limit the home-range size of the largest groups by affecting their speed or their tendency to move to new areas. As a result of the interaction between these two effects, we expect that there will be a non-linear relationship between group size and movement, which, if it results in more effective use of available resources, could correspond to an optimal group size for group movement characteristics.

We predict that intermediate groups will have the largest home-range sizes. This will arise because while larger groups should travel further, intermediate groups will have lower day-to-day home-range fidelity. We further predict that intermediate size groups will present the fastest speed while 'on the go'. However, these factors may be tempered by group composition. Groups in our population reproduced several times over our study period, and the presence of young group members could restrict the movements of groups. Thus, we run our analysis while controlling for the number of chicks present in the groups, and predict that groups with more chicks should travel slower and shorter distances, thereby restricting their home-range size. Finally, we predict that a non-linear relationship between group size and movement should translate to corresponding peak in fitness, measured as the number of chicks in a group. Such a pattern would suggest that maximising the difference between navigation and coordination represents an optimal group size at which groups are most efficient in exploiting their environment.

## Results

Our movement analysis, based on 1,274,434 five-min GPS positions, spanned 45 seasonal observations of individuals (groups) over five 2-month-long seasons (*Figure 1*, see *Supplementary file 1A*

for a summary of the data). Group sizes, recorded during daily census observations, were widely distributed, with intermediate groups being observed less often (*Figure 2*).

Fitting generalised estimating equations (GEEs) with the number of adults (group size) and the number of chicks as predictors (while accounting for repeated measures of the same groups across seasons) revealed that home-range size was significantly predicted by group size and its quadratic term, but that this relationship was modulated by the presence of chicks. Intermediate groups had the largest home-range, but home-range size decreased with increasing number of chicks (*Table 1A*, see *Supplementary file 1B* for all model fits). Average temporal overlap in space use was also significantly predicted by group size and its quadratic term, but was not significantly modulated by the presence of chicks (*Table 1B*, see *Supplementary file 1C* for all model fits). Average daily distance travelled increased linearly with group size (*Table 1C*, see *Supplementary file 1D* for all model fits). Speed 'on-the-go' was not explained by group size or by number of chicks in the group (*Table 1D*, see *Supplementary file 1E* for all model fits). Overall, smaller and larger groups expressed smaller home-range sizes and increased average temporal overlap in space use, while larger groups expressed longer daily travel distances than groups of smaller size (see *Figure 3* and *Figure 4*).

The number of chicks in the group was also significantly predicted by group size and its quadratic term (*Table 1E*, *Figure 3D*, see also *Supplementary file 1F* for all model fits). We found that intermediate-sized groups had more chicks, with the peak in fitness (group size = 33) closely matching the peak in home-range size (group size = 36) and average daily overlap (group size = 37).

Number of adults (group size) and its quadratic term significantly predicted (A) home-range size and (B) average temporal overlap in space use. The presence of chicks modulated the relationship between group size and home-range size, reducing the overall contribution of the number of adults to home-range size (the negative interaction with number of adults) and reducing the quadratic effect (the positive interaction with number of adults squared, which offsets the negative quadratic coefficient). (C) The average daily distance travelled increased with number of adults (see *Figure 3C*). (D) Speed while travelling was not significantly impacted by group size or its quadratic term. (E) The number of chicks per group was significantly predicted by group size and its quadratic term (see *Figure 3D*). Parameter estimates, significance, and $R^2$ were generated using the most parsimonious fitted GEEs based on QIC (see Materials and methods). All models tested are summarized in the *Supplementary file 1B to 1F*.

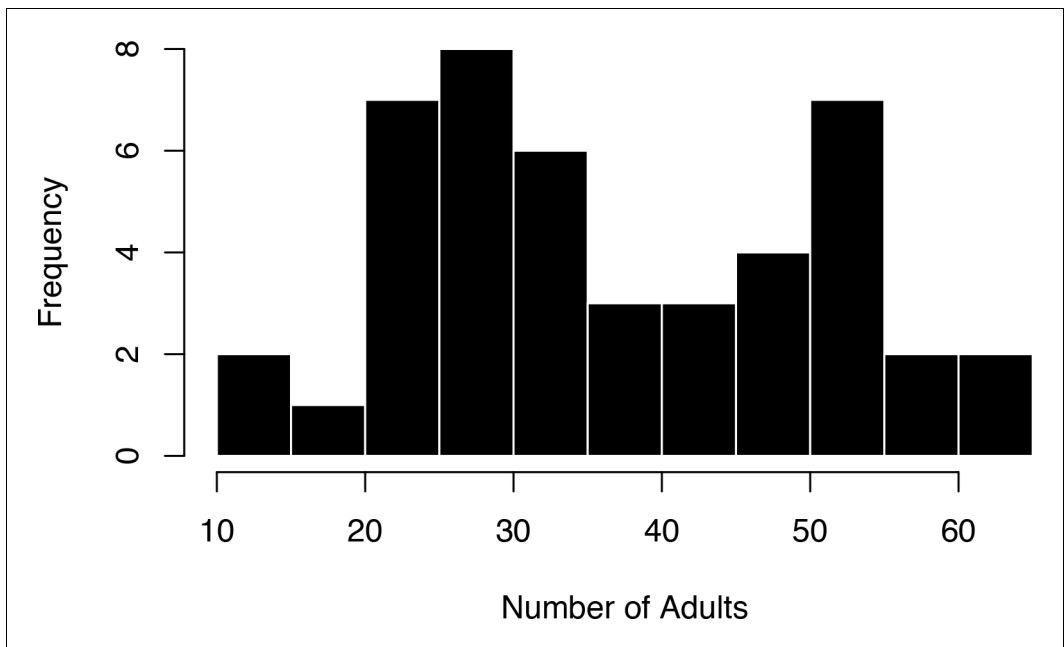

**Figure 2.** The distribution of number of adults in each of the focal groups is bimodal. The most common group sizes were either smaller or larger, with fewer groups having an intermediate group size.

**Table 1.** Summary of the GEE results.

| Response variable | Model ID | Independent variables | Estimate | Standard error | Wald | P | $R^2$ | QIC |
|---|---|---|---|---|---|---|---|---|
| A  Home range | m18 | Intercept | −1.37 | 1.12 | 1.50 | 0.221 | 0.187 | 60 |
| | | Number of Adults | 0.224 | 0.08 | 7.86 | **0.005** | | |
| | | Number of Adults$^2$ | −0.003 | 0.001 | 8.42 | **0.004** | | |
| | | Number of Adults:Number of Chicks | −0.006 | 0.002 | 7.91 | **0.005** | | |
| | | Number of Adults$^2$:Number of Chicks | $1.225 \times 10^{-4}$ | $4.93 \times 10^{-5}$ | 6.16 | **0.013** | | |
| B  Average temporal overlap in space use | m6 | Intercept | 1.025 | 0.10 | 101.31 | **<0.001** | 0.098 | 3.58 |
| | | Number of Adults | −0.013 | 0.01 | 4.64 | **0.031** | | |
| | | Number of Adults$^2$ | $1.744 \times 10^{-4}$ | $7.577 \times 10^{-5}$ | 5.30 | **0.021** | | |
| C  Average daily distance travelled | m11 | Intercept | 6.068 | 0.68 | 79.97 | **<0.001** | 0.247 | 116 |
| | | Number of Adults | 0.067 | 0.02 | 16.51 | **<0.001** | | |
| D  Speed while Travelling (m/s) | m14 | Intercept | 0.143 | 0.03 | 21.70 | **<0.001** | 0.191 | 3.78 |
| | | Number of Adults | 0.002 | $2 \times 10^{-3}$ | 1.43 | 0.232 | | |
| | | Number of Adults$^2$ | $1.602 \times 10^{-5}$ | $2.73 \times 10^{-5}$ | 0.35 | 0.556 | | |
| E  Number of chicks | m73 | Intercept | −6.491 | 7.04 | 0.85 | 0.357 | 0.123 | 1295 |
| | | Number of Adults | 1.047 | 0.49 | 4.65 | **0.031** | | |
| | | Number of Adults$^2$ | −0.016 | 0.01 | 4.76 | **0.029** | | |

## Discussion

Our study shows that group size and composition can impact collective movement characteristics. Groups of vulturine guineafowl that have an intermediate size exhibit larger home-ranges and explore more new areas day-by-day. However, this relationship exists despite intermediate groups travelling shorter distances each day compared to larger groups. Even though the presence of chicks does not impact how far or how fast groups travel, having many chicks restricts the home range of their group. Further, our study also shows that groups of intermediate size have higher fitness as they produce more offspring than larger or smaller groups. Our results are therefore consistent with our a priori prediction that a non-linear relationship should exist between group size and collective movement, reflecting the increasing navigation ability with increasing group size and the increasing challenges of coordinating movement in larger groups. The relationship between group size and reproductive output further suggests that intermediate-sized groups are more effective at exploiting space, providing evidence for an optimal group size for collective movement.

The relationships between group size and group movement outcomes provide new insights into how differently sized animal groups interact with their environment. Larger groups cover larger distances during the day, which is likely to be important for encountering sufficient resources to meet their energetic demands, as proposed by the ecological constraints model (*Chapman and Chapman, 2000*; *Ganas and Robbins, 2005*; *Gillespie and Chapman, 2001*; *Teichroeb and Sicotte, 2009*). However, by exploiting larger areas over a whole season, groups of intermediate size are likely to benefit from having access to a greater abundance and/or diversity of food resources (*Maher and Burger, 2011*), they potentially also benefit both from being less predictable to predators (*Richardson et al., 2018*; *Roth and Vetter, 2008*), and from having access to a more diverse social landscape. In addition, intermediate-sized groups achieve larger home ranges while travelling shorter distances than larger groups, implying that they have lower energy expenditure. It is worth noting that we focused our analysis on 2-month periods each capturing similar ecological conditions. Future studies could investigate whether these non-linear relationships between group-size and movement characteristics change across different environmental conditions and especially in seasons where energy might be under tighter constraints, such as during droughts. Evidence for this could bring important insights into the drivers of group size variation.

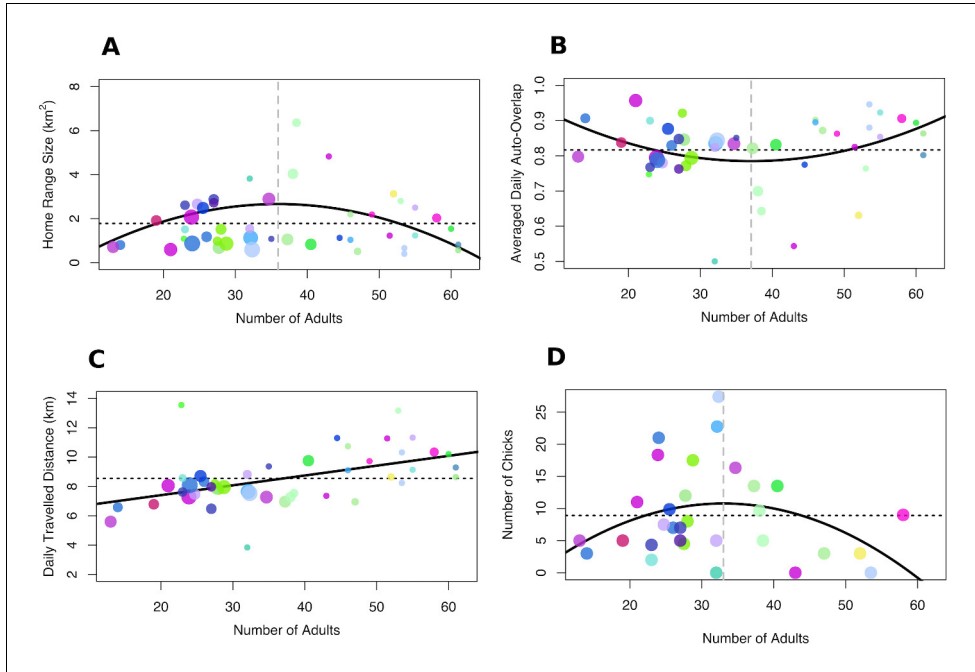

**Figure 3.** Relationship between group size, collective movement parameters and fitness. (**A**) Home-range size and (**B**) average daily home-range overlap were maximised and minimised at intermediate group sizes, respectively. By contrast, (**C**) average daily distance travelled increased with group size. (**D**) Fitness, measured as the number of chicks per group in seasons when chicks were present in the population, was maximised for intermediate-sized groups. Each dot represents one group in one season and the colours for each group remain consistent with *Figure 1*. The size of the dots represents the number of chicks in the group for panels A–C. Solid black lines represent the fit of the data (in the absence of chicks for panels A–C), dashed black lines represent the mean across all group sizes and vertical dashed grey lines in A, B, and D represent the group size for which the curve of the response variable reaches a maximum (A and D) or a minimum (B).

While we predicted that group size would translate to differences in movement while 'on-the-go', our measure of moment-by-moment movement (speed while travelling) did not reveal any relationship between group size and movement speed. One reason for this could be that larger groups could have better knowledge of the environment, meaning that they can offset consensus costs (i.e. periods of slow movement due to conflict within the group regarding movement directions) with more effective movement once departed (*Couzin et al., 2005*). For example, the speed and polarization, and subsequently the directionality of group motion, increases with the size of fish schools (*Tunstrøm et al., 2013*). As group size increases in locusts, they transition from disordered to ordered motion (*Buhl et al., 2006*). Thus, it is possible that while the realised movement speed does not differ with group size, the characteristics of how the movement is achieved could vary between smaller and larger groups. Alternatively, the challenge of reaching a consensus could mean that larger groups end up repeating the same routes more often, which they can achieve by moving faster as individuals all know where they are going. This explanation is consistent with our findings that larger groups were less variable in where they travelled each day. Such mechanisms linking collective movement with ranging behaviour warrant further investigation.

Several studies in the past have investigated the relationship between group size and ranging behaviour. Most notably, *Markham et al., 2015* found that intermediate-sized groups of wild baboons (*Papio cynocephalus*) utilized smaller areas, covered shorter distances per day, and used space less evenly than smaller and larger groups—patterns which are largely opposite to our findings. *Stevenson and Castellanos, 2000* found that woolly monkeys (*Lagothrix lagothricha*) in groups of intermediate size travelled less during the day than larger and smaller groups. Finally, research on meerkats (*Suricata suricatta*) *Bateman et al., 2015* found no relationship between group size and group movement characteristics. However, in most previously studied systems, group movement characteristics are likely to be impacted not only by the ability for groups to move, but also by intra-

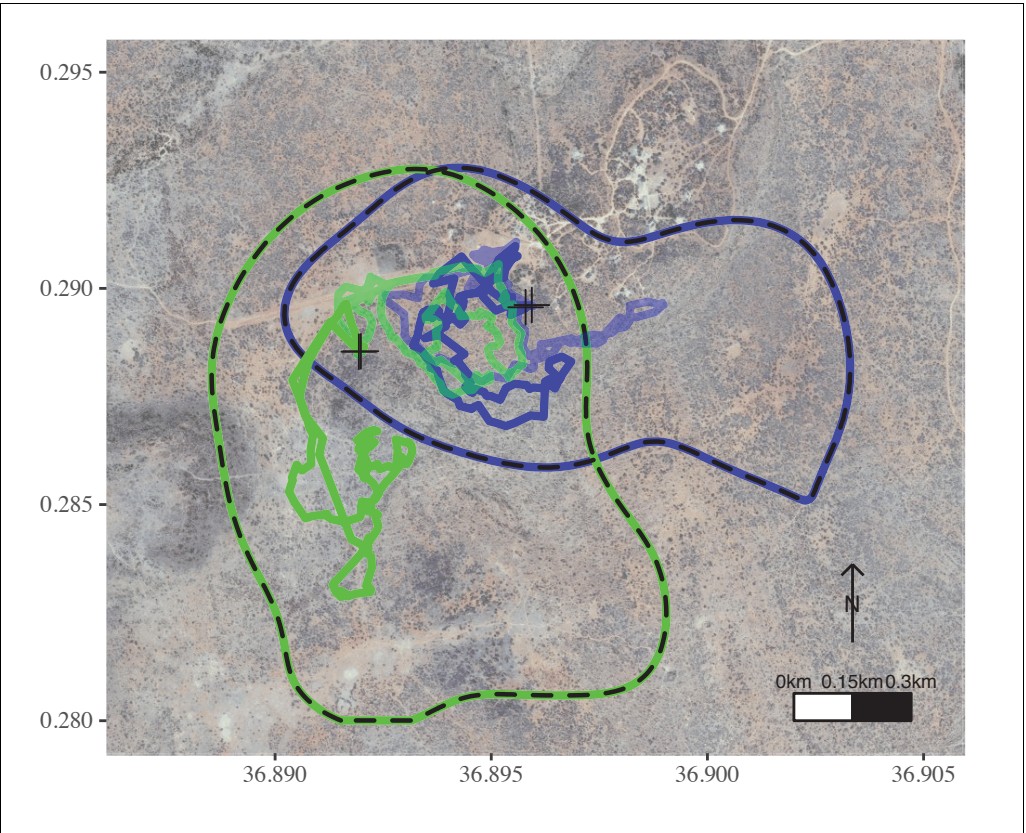

**Figure 4.** Movement characteristics of intermediate-sized groups differ from those of large groups. In a given season, a group of intermediate size (green) ranged in a wider area than a larger group (blue) and explored different areas on consecutive days, while the two groups overlapped at the same areas on the same days. Neither of these groups had chicks. The dashed lines represent the 95% AKDE for Season D for each focal group. The transparent and non-transparent tracks are from 2 consecutive days. Crosses represent the night roosts for these 2 consecutive days. Groups are coloured as in *Figures 1* and *3*.

group contest competition and inter-group competition (*Markham et al., 2015*). For example, smaller groups have the weakest resource-holding potential, meaning that they should have the smallest territories. At the same time, they might be excluded from the best, most resource-rich, habitats by larger groups (who need more resources due to intra-group competition), meaning that they could also end up having to use equally sized or even larger areas (*Markham et al., 2015*; *Stevenson and Castellanos, 2000*). In contrast to previous studies, our evidence for a non-linear relationship between group size and movement characteristics comes from a species that not only presents minimum intergroup competition but also groups use largely overlapping areas. These features reduce the confounds on the link between group size and group movement behaviours. Our study is therefore more likely to reveal effects arising from movement-based mechanisms without confounding effects that are exogenous to the group.

Vulturine guineafowl live in a multilevel society, where groups preferentially associate with other groups (*Papageorgiou et al., 2019*). In such societies, intergroup associations can bring many important benefits, such as information sharing on predators and resources, as well as mating and dispersal opportunities (*Grueter et al., 2020*). A large home range size in a multilevel society could provide groups with access to more communal roosts (we have observed up to five groups in a single roost, *Papageorgiou et al., 2019*) and more glades, and thus more opportunities to associate with more groups. However, the benefits of associating with other groups might not increase linearly with home range size, as associating with a large number of conspecifics can also imply costs, such as disease transmission or increased competitions for mates (*Krause and Ruxton, 2002*). More work

is needed on the interactions between ranging behaviour of groups and the broader social environment in multilevel societies.

We built on existing insights from studies of collective behaviour to develop an a priori prediction that the relationship between group size and movement characteristic should be non-linear. Our prediction proposed that maximising the use of space (i.e. a larger home range) through effective movement (faster movement and less repetitive ranging) should result in more efficient exploitation of the resources available to a group. A key question is whether groups actually benefit from this maximisation? By analysing data on the reproductive success across the groups, we found striking evidence for a peak in fitness, with intermediate-sized groups having more chicks. This peak, which closely matches the group size peaks in movement characteristics (home range size and average daily overlap), suggests that intermediate group sizes might be linked to higher reproductive success. Our findings therefore provide evidence for an optimal group size in the context of collective movement.

An interesting additional finding from our study is that optimally sized groups (33–37 individuals) were relatively rare. Instead, the distribution of group size appears to be bimodal with most groups being either smaller or larger than the intermediate group size. This observation aligns with the hypothesis that while an optimal group size results in maximum individual fitness (*Pride, 2005*) groups at the optimal size should be unstable and therefore rarely observed (*Sibly, 1983*). Groups in our study grow either through reproduction or through immigration. We found that intermediate-sized groups have high reproductive success, which naturally pushes them beyond the optimal. Further, because dispersing individuals should prefer optimal size groups, they should also turn intermediate-sized groups into groups that are larger than the optimal size. Future studies should examine whether groups near the optimal size for collective movement also highly attract more dispersers.

The observed variation in results between our study and previous studies investigating the effect of group size on ranging behaviour in mammals could also potentially arise due to methodological differences. For example, we used high-resolution tracks and continuous-time movement modelling to get much more precise estimates of movement characteristics. By contrast, other studies have used much lower sampling resolutions, often relying on hand-held GPS devices (*Bateman et al., 2015*; *Markham et al., 2015*) or used simplistic minimum convex polygons to model home-ranges (*Markham et al., 2015*). Minimum convex polygons are unlikely to be very good estimates of home-ranges as they are sensitive to outliers (e.g. from transcription errors or poor GPS fixes) and capture only one element of space use (the extreme boundaries, which could be influenced by a range of factors such as habitat geometry; *He et al., 2019*). However, despite these potential methodological differences, we believe that conflicting findings are more likely to suggest that the effect of group size on ranging behaviour and collective movement varies according to the properties of the different animal social systems. For example, we predict that the outcome of the many factors that contribute to the ranging characteristics of social groups will be modulated by the distribution of food resources each species exploits. Vulturine guineafowl have very little need to defend areas from other groups or compete for resources within groups, as their food source (insects and small seeds) are widespread and they roost on the top of dense stands of trees, a habitat feature which is common in the study area. When competition is decreased, we predict that group size could become a more important driver of collective movement outcomes. We hope that by collecting more data on this question across species there will soon be sufficient studies to fully test this prediction.

In addition to the effects of the number of adults in the group, we also revealed the impact of having young present in the group. Vulturine guineafowl are precocial, meaning that they are very small when they join the group, and mature over long periods of time—at 1 year of age they are only approximately 65% of the full adult size. For a number of months after hatching, chicks are unlikely to contribute to making decisions, but are highly vulnerable to predators—causing them to remain more in cover. At the same time, chicks are likely to move more slowly, or cover less ground, due to their smaller body size. We found that having chicks in the group has a profound effect on home-range size. Our results indicate that groups with more chicks use smaller areas, thereby limiting groups of intermediate sizes from exhibiting larger home-ranges. The limitations that chicks have on group movements raise interesting questions about the costs of group living. The fact that groups of vulturine guineafowl remain cohesive across seasons and years suggests that any costs borne from collective movement will be shared by all group members, including by individuals which have failed to reproduce (vulturine guineafowl are plural breeders; *Papageorgiou et al., 2019*) or by

subadults who have yet to disperse. Thus, our data reveals a potentially important source of conflict among group members. Such conflicts may be common in plural-breeding groups, which are commonly found in multilevel societies.

We have shown that using synchronous and high-resolution GPS tracking of multiple social groups that reside in the same area can reveal new insights into the movement and social ecology of species. Specifically, we demonstrated that group size and composition can predict collective movement and space use characteristics. Our results are consistent with the prediction that increased navigation accuracy but decreased coordination ability could produce a non-linear relationship between group size and collective movement. However, the mechanisms that underlie the interaction between coordination maintenance and navigation accuracy in free-living groups of various sizes remain to be discovered. Further, we have identified dimensions of movement—specifically that intermediate-sized groups have larger home-ranges and use a larger diversity of areas—that translate into intermediate-sized groups having greater efficacy at exploiting their environment, and, therefore, higher fitness. These results suggest the presence of an optimal group size for collective movement. A further interesting observation is that the benefits of having an optimal group size are not present when groups have offspring, and that few groups are of an optimal size, highlighting a potentially important feedback between current and future reproductive success. The ability to develop and test our novel hypotheses relating to collective movement of animals in the wild was only possible by conducting a large-scale multi-group tracking study in a species that does exhibit little intra-group and inter-group competition. We hope that cheaper, more reliable, and smaller devices, will continue to reveal similar insights into the lives of wild animals across the diversity of social systems.

## Materials and methods

### Study site and study species

We studied a population of vulturine guineafowl that resides in an area of approximately 500 ha in the southern part of the Mpala Research Conservancy (MRC) in Laikipia, Kenya. Vulturine guineafowl are large, predominantly terrestrial, and highly gregarious birds. They primarily forage on the ground, feeding on small grass root buds, seeds, fresh grass and small invertebrates, in other words widely dispersed food sources that have been predicted to limit within- and between-group contest competition (*Dammhahn and Kappeler, 2009*). Based on 3 years of observations of colour-marked individuals, we found that they live in social groups, which range in size from 13 to 65 adults, have stable membership across years and remain completely cohesive (with individuals rarely out of sight of other group members) during non-breeding seasons (typically January to March and July to October) (*Papageorgiou et al., 2019*). Within a local area, groups form a multilevel society, which includes groups roosting communally (with roosts containing several hundred birds) and groups regularly encountering other groups during the day (at which times between-groups agonistic interactions are rarely observed) (*Papageorgiou et al., 2019*). Intergroup encounters and foraging are most common on glades (*Papageorgiou et al., 2019*), which are small deforested areas that were previously used as overnight shelters for cattle and are therefore are rich in nutrients (*Young et al., 1995*). All vulturine guineafowl groups have access to glades as they are numerous in our study area. On glades, vulturine guineafowl are prone to predation by raptors, such as martial eagles (*Naude et al., 2019*), while moving through dense vegetation makes them prone to ambush by carnivorous mammals, such as jackals or leopards.

### Data collection

We trapped nearly all the adult members from each of the groups in our study area using large walk-in traps. All trapped individuals were kept inside covered cages (1 m*50 cm*50 cm cage with 2 cm plastic mesh covered by a canvas) to minimize stress and allow birds to remain standing while awaiting processing. Birds were removed one by one, measured, and ringed with an individually numbered National Museums of Kenya stainless steel ring (fitted on the tarsus of the right leg) and a unique combination of four plastic colour bands fitted on each leg, on the tibiotarsus, for field identification. For more details on trapping and marking processes, see *Papageorgiou et al., 2019*. In each group, we fitted high-resolution solar-powered GPS tags (15 g Bird Solar, e-obs Digital

Telemetry, Grünwald, Germany) to between one and five individuals. The devices were elevated using neoprene pads to prevent feathers from covering the tag's solar panel, and were attached to the body using a backpack harness design (Teflon ribbon), which included a cotton thread safety release mechanism. GPS tags stayed on the birds from 6 months to 2 years, before the cotton breaks. Groups were trapped once per year to mark new members and check the condition of the birds which carry GPS tags. Since the start of the study in 2016, no bird has ever been found with problems due to the GPS tags. Each tag (tag, pads, ribbons, crimps) weighed approximately 20.5 g, far less than the recommended 3% of the birds' body weight (*Kenward, 2001*). Each device was programmed to record one data point (date, time, coordinates) every second when the battery had a high charge (approximately every 2nd to 3rd day, for up to 8 hr continuously). When the battery was at the next lower threshold, we set tags to record one point per second for the first 10 s of every 5th minute. If battery charge was very low (this setting was rarely used during our study period), tags were set to record one point every 15 min. Data were remotely downloaded every 2 or 3 days using a BaseStation II (e-obs Digital Telemetry, Grünwald, Germany). Field-testing suggested that that the relative spacing of tags was accurate to within 1 m, 95% of the time tracked. We uploaded all of our data to the Movebank repository (*Wikelski and Kays, 2018*), see https://www.movebank.org/cms/webapp?gwt_fragment=page=studies,path=study475851705 for further details.

## Study seasons

We selected seasons with intermediate rainfall and greenness, as described in *Papageorgiou et al., 2019*. Vulturine guineafowl groups temporarily split during the breeding (wet) season as breeding pairs form, subadults disperse from their natal groups and females lay and incubate their eggs (*Papageorgiou et al., 2019*). It is therefore challenging to quantify membership accurately during the breeding season. Breeding seasons can occur twice a year (during April or May and during October or November).

We also excluded periods of drought, as these cause groups to regularly travel to large bodies of water, such as the river that crosses our study area. This unusual situation (guineafowl are adapted to rarely require access to standing water under normal conditions) would make the estimation of home-range size confounded by how far from the river each group roosts. We thus made our selection of seasons based on the seasonal conditions where groups remain cohesive but do not make extraordinary movements, while remaining blind to the specific movement patterns of the GPS-tagged birds. We defined seasons as being 2 months, which corresponds to period changes in conditions at our study site.

Given that our entire study population inhabits a small area, approximately 500 ha, and that all groups can easily access all parts of our study area, we assume that groups' relative ranging behaviour was not restricted by the environment during these intermediate seasons. Data for this study were collected from May 2017 to February 2020, and the specific time periods that fulfilled the criteria listed above were categorised as Season A (30.5.2017 to 1.8.2017), Season B (15.12.2017 to 15.2.2018), Season C (1.8.2018 to 29.9.2018), Season D (10.6.2019 to 9.8.2019), and Season E (14.1.2020 to 16.3.2020).

## Census data

Census data was collected daily from a vehicle driving along the roads in our study area. Every time a group was encountered, the identities of the marked birds were recorded, and a total number of birds present was counted (including unmarked birds). Each observation of a 'group' is defined as being a single or a multiple group, depending on whether the group moves away cohesively or whether individuals move away in different directions. Counts are split into the number of adults and the number of chicks, with chicks being easily distinguishable until ~ 9 months of age.

## Study group selection

Within each season, we selected groups containing one or more GPS tagged individuals for which we had precise group-size counts where observers could clearly see and count all the group members being present (i.e. groups were counted more than once by more than one observer, the group size was found consistent across counts). We included data on 21 different observational datasets of groups, with some groups replicated in more than one of the seasons (*Supplementary file 1A*,

*Figure 1*). For seasons in which a group had chicks, we calculated the median number of chicks per group, across all observations, in each study season.

## Movement analysis

We used the GPS data from each group in each season to quantify movement behaviour and resulting home-range characteristics. In groups containing more than one GPS-tagged individual, we randomly selected one to represent that group's movement, given that groups move very cohesively (typically group members are within 29 m of each other 95% of the time tracked; *Papageorgiou et al., 2019*). We calculated (1) the core home-range size (50% auto-correlated kernel density estimate), (2) average temporal overlap in space use, (3) average daily distance travelled. We also explored (4) speed while moving. We used the built-in features from the Movebank data repository to remove the very rare outliers in our dataset that were falling outside Kenya. For (1 , 2) and (3) we used data points collected on the tenth second of every fifth minute, sub-setting this from high-resolution (1 Hz) data and from data when the tag was set to collect 10 points every 5th minute. This approach allowed us to ensure that the data points from all GPS-tagged birds were precisely synchronised in time.

For core home-range size (1), we calculated home-range sizes by fitting a continuous-time movement model (ctmm) to each individual GPS dataset (one bird per group per season). These models follow a continuous-time stochastic process from which the maximum likelihood Gaussian home-range area can be extracted after the best fitted model is selected based on AIC. Because ctmms model the actual movement, they better predict home-ranges than classical approaches, such as minimum convex polygons, that are typically prone to being highly influenced by just a few data points (*Fleming et al., 2015*). We used the maximum likelihood home-ranges to determine the area in which each bird was located 50% of the time. This procedure was done using the 'ctmm' R package (*Calabrese et al., 2016*; *Fleming and Calabrese, 2018*).

For average temporal overlap in space use (2), we measured the inverse of the exploration of new areas, for each bird in each season. We first fitted a ctmm to the 5-min data from each bird for the first 21 days it was tracked in each season (longer time spans were not computationally feasible for this particular analysis). We then calculated the overlap of the daily auto-correlated kernel density estimation for home-range of each bird, using the 'overlap' function of the 'ctmm' R package (*Winner et al., 2018*). Specifically, we calculated the overlap between daily ranges that were up to 3 days apart. Finally we calculated the mean of the spatial overlap of one group with itself across each consecutive sets of 3 days, for the entire 21-day period, for each group.

For average daily distance travelled (3), we used the 5-min data from above (one bird per group per season) and the function 'speed' from the ctmm R package, which provides the mean travel distance per day but not the speed while 'on the go'.

To estimate the mean speed 'on the go' (4) for each individual in each season we used only a subset of the data representing the times when the GPS tags were collecting one point per second (1 Hz). From these data, we created a histogram of the log of the movement speeds (distance travelled between consecutive points), revealing that speeds were multimodally distributed for each individual. We found that the minima between the second and the third mode corresponded to $-3.5$ (or $0.03 \text{ ms}^{-1}$) and classified speeds exceeding this value as 'moving' and speeds slower than this value as 'stationary' (*Bender et al., 2011*; *Wilson et al., 2015*). We then used the data classified as 'moving' to calculate the mean of the distance travelled in consecutive seconds of the 1 Hz data.

## Statistical analysis

We fitted GEEs using the R package 'geepack' to test whether group size and number of chicks predicted home-range size, average temporal overlap in space use, average daily distance travelled and speed 'on the go'. Our primary predictor variable was the number of adults in the group. Because chicks could modulate the relationship between the number of adults (our measure of group size) and movement response variables, we also allowed models to include a term for the number of chicks. Because we expected a non-linear relationship between group size and movement parameters, we also allowed models to include the number of adults squared (the quadratic term). Further, because we did not know how chicks could modulate group movement, we allowed the number of chicks and the number of chicks squared to be added in interactions with the number of adult and

the number of adults squared. We created all combinations of response variables and interactions (only between terms containing adults and chicks), but ensuring that models contained at least the number of adults as a predictor, and always included the number of chicks when the number of chicks squared was included. We used QIC to select the most parsimonious models (lowest QIC) (*Hocking, 2014*) and we also calculated the $R^2$ of each GEE to estimate their goodness of fit (*Zheng, 2000*). For all models, we fitted group ID as a repeated measure to account for multiple measures of the same group across seasons. All calculations and statistical tests were conducted in R version 3.5.1 (*R Development Core Team, 2018*) with the significance threshold set at $p \leq 0.05$.

## Group size and fitness

We investigated the relationship between group size and reproductive success by fitting a GEE containing group size and it's quadratic term as predictors of the median number of chicks in the group in a given season. We limited this analysis to seasons during which there were chicks in our study population. In Seasons A and D, the study population had no chicks since they had not bred for more than 10 months. During the rest of the three study seasons (Seasons B, C and E), the chicks where between 3 and 5 months old, and each season represented a different cohort of chicks.

## Data accessibility

The code to run the ctmms and conduct the statistics is available on https://github.com/DanPapageorgiou/Group_size; *Papageorgiou, 2020*; copy archived at swh:1:dir: fcd6639f112b1d96bfe8d122f96a2c0ed2a1673b.All data used in this study are stored on Movebank under the study name Avulturinum_Farine: https://www.movebank.org/cms/webapp?gwt_fragment=page=studies,path=study475851705.

## Acknowledgements

We thank the Mpala Research Centre, and the Ornithological Section of the National Museums of Kenya for supporting this research work. We also thank Lucy Aplin and Stephen Lang for useful discussions and comments on the manuscript. We are grateful to John Ewoi, Brendah Nyaguthii, Alex Baiywa, Wismer Cherono, Charlotte Christensen, James Klarevas-Irby and Sylvester Karimi for field assistance. All work was conducted under research permits from the Max Planck Society Ethikrat Committee (2016_13/1), the National Commission for Science, Technology and Innovation of Kenya (NACOSTI/P/16/3706/6465) and under a Research Authorisation and a Capture Permit by the Kenyan Wildlife Service.

## Additional information

### Funding

| Funder | Grant reference number | Author |
|---|---|---|
| Max Planck Society | | Danai Papageorgiou<br>Damien Roger Farine |
| Daimler und Benz Stiftung | 32-03/16 | Damien Roger Farine |
| Association for the Study of Animal Behaviour | | Damien Roger Farine |
| Horizon 2020 - Research and Innovation Framework Programme | grant agreement No. 850859 | Damien Roger Farine |
| DAAD | | Danai Papageorgiou |
| National Geographic Society | Early Career Grant WW-175ER-17 | Danai Papageorgiou |
| International Max Planck Research School for Organismal Biology | | Danai Papageorgiou |
| Deutsche Forschungsge- | Centre of Excellence 2117 | Danai Papageorgiou |

meinschaft                    "Centre for the Advanced     Damien Roger Farine
                              Study of Collective
                              Behaviour" ID: 422037984

The funders had no role in study design, data collection and interpretation, or the decision to submit the work for publication.

## Author contributions
Danai Papageorgiou, Conceptualization, Data curation, Formal analysis, Funding acquisition, Investigation, Visualization, Methodology, Writing - original draft, Project administration, Writing - review and editing; Damien Roger Farine, Conceptualization, Formal analysis, Supervision, Funding acquisition, Methodology, Writing - review and editing

## Author ORCIDs
Danai Papageorgiou (iD) https://orcid.org/0000-0001-5535-6133
Damien Roger Farine (iD) https://orcid.org/0000-0003-2208-7613

## Decision letter and Author response
Decision letter https://doi.org/10.7554/eLife.59902.sa1
Author response https://doi.org/10.7554/eLife.59902.sa2

## Additional files

### Supplementary files
• Supplementary file 1. Summary data and GEEs. (A) includes the summary data for each group across the five study seasons. (B-F) present all GEEs fitted. Those with the lowest QIC, for each response variable, are also presented in the main text.

• Transparent reporting form

### Data availability
The code to run the ctmms and conduct the statistics is available on https://github.com/DanPapageorgiou/Group_size (copy archived at https://archive.softwareheritage.org/swh:1:dir:fcd6639f112b1d96bfe8d122f96a2c0ed2a1673b). All data used in this study are stored on Movebank under the study name Avulturinum_Farine: https://www.movebank.org/cms/webapp?gwt_fragment=page=studies,path=study475851705.

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
