## [Decision Letter]

**Acceptance summary:**

This paper combines an impressive continuous movement dataset, with demographic and social data. The addition of the chick data really does help to complete the manuscript by providing support for the argument that an intermediate group size is beneficial. This type of information is very rare and delivers invaluable insights into movement ecology.

**Decision letter after peer review:**

Thank you for submitting your article "Group size and composition influence collective movement in a highly social terrestrial bird" for consideration by *eLife*. Your article has been reviewed by three peer reviewers, including Samuel L Díaz-Muñoz as the Reviewing Editor and Reviewer #3, and the evaluation has been overseen by Christian Rutz as the Senior Editor.

The reviewers have discussed the reviews with one another and the Reviewing Editor has drafted this decision to help you prepare a revised submission.

Summary:

This paper is an empirical investigation into the question of optimal sizes and how group size affects or influences collective movement. The study uses a very interesting study system, focusing on a highly social terrestrial bird that is non-territorial, is found in a range of group sizes, and appears to have high social cohesion within those groups. The study employs an impressive dataset of continuous movement tracking technology, together with demographic and social data. This type of dataset is very rare in the field and represents a treasure of potential insights. The text reports that intermediate sized groups had the largest home range size and variation in site use. Additionally, the presence of young individuals resulted in smaller ranges compared to groups with not young individuals. These results indicate that the size and composition of groups is an important factor shaping space-use patterns of social species. Based on these results, the manuscript concludes there is an optimal group size for movement.

In preparing your response and manuscript modifications, please pay special attention to the Essential revisions section below. The majority of these pertain to the framing of the paper, predictions, and conclusions, which are not supported by the data, particularly with regard to the concept of optimality and connections to fitness. The revisions should either modify the conclusions made or employ some data (some of which are included in the manuscript) to support them.

Essential revisions:

1) The main revision regards the conclusions drawn in the paper from the data. In part, there are two interacting issues: the framing of the paper and the data support for the optimality conclusion. The theoretical premise of optimal movement does not seem supported in general, and no specific support for the study species is provided. Specifically, the assumption that larger home ranges are better is not well supported and contradicted in some parts of the text. The other predictions (greater variability day-to-day and faster on-the-go movement) could be plausibly argued either way, and no support for the predictions specifically with regard to guinea fowl is presented. Interestingly optimality of fitness (which is mentioned with no support in the text at several points) could actually be assessed using proxies of fitness, such as number of chicks, which are available in the dataset but not used for this purpose in the analyses.

Please see individual comments for potential resolutions below, but either a restating of the conclusions or a re-analysis of existing data to support conclusions is needed.

1a) The blanket statement that larger home ranges are better is not justified. Large territories or ranging can be signs that things are going wrong (ranging increased for scarce mates, or because resource shortage). This is acknowledged explicitly in the text about the study species, indicating that birds will travel far distances to gain access to the river in time of drought (subsection “Study seasons”). The evidence provided for larger home ranges being better mentions that Florida scrub jay groups that produce more offspring have larger territories (Introduction, third paragraph). Do they maintain larger territories because they are better groups? This question is actually partially answered in the fifth paragraph of the Introduction. The end result is all these statements undermine the key prediction and conclusions of the paper.

1b) Bottom line is that the question "Optimal for what?" should be answered. There is clearly an intermediate pattern, but that doesn't necessarily indicate optimality. If we think of optimal foraging theory for instance, there are clearly curved distributions say, in handling time or movement, but what makes them optimal, say for energy, is that there is a measure (usually calories) that coincides with that intermediate peak. That is what is being optimized (energy) is clear, can be objectively measured (calories), and has a clear effect on body condition or fitness. In this case, it is not clear what the movement is being optimized *for*. A potential way to resolve this issue is to provide some evidence that the factors affecting home range size mentioned in the Introduction actually affect guinea fowl. Is predation indeed higher in birds with more predictable routes? Are larger home range sizes measurably better in terms of resources, bird condition, offspring production, etc? Ultimately, if the desire is to connect movement to fitness (the ultimate currency, in evolutionary terms) then you could measure number of chicks as a measure and these data seem to be available, which is indeed rare data to have! The more I think of it, the more it seems like a lost opportunity, especially when there are many mentions of fitness and movement relating to fitness, but no data to support the linkage (e.g. Discussion, seventh paragraph). Otherwise, the conclusions should be limited to how the size/composition of groups shapes space use.

1c) Optimal group sizes: If intermediate sizes are optimal, why should you expect to observe wild groups to be smaller or larger than the optimal? There's a nice description of why group sizes that are unstable at the end of the paper (Discussion, seventh paragraph) – moving some of this description to the Introduction would help set up the ideas tested in the study more strongly (rather than waiting until the end of the paper). Related to this point: Why is Figure 3 in the Discussion? This should be a major part of the paper given the title and it is rarely appropriate to have figures presenting primary data in the Discussion.

1d) Why was speed on the go only presented in supplementary material? Even if they don't support the predictions they should be in the main text. Actually, especially if they don't support the predictions.

1e) The data on chicks slowing down movement is very important and not addressed enough in the text. Together with the data on chick productivity and group size distributions (which for some odd reason is in the Discussion) there could be really interesting insights on questions that are not the focus of this paper for which the data exists to test.

2) The reviewers had a number of questions regarding potential confounds in the study or factors to control for. These are very important to verify, although I suspect many of these can be addressed rather quickly. Note that some of these points (e.g. group success) have a bearing on larger issues in the manuscript raised above. In any case, they should all be reported on in the final manuscript.

2a) Did the number of individuals tagged in a group have any impact on the results? Was this accounted for in the models?

2b) There may have been different environmental conditions across the different years that might have influenced the group movements. Did you account for the different years or for the 5 different seasons in your models, such as a random effect?

2c) Figure 1: Are all 21 groups represented in each of the 5 seasons? Looking at Season A, there seem to be only 5 groups. If there are a different number of groups per season, do you pool the data for the analyses? If so, do you account for the different number of samples across the different groups, such as weighting the values?

2d) Treatment of chicks: The authors find that the presence of chicks affects group movements. However, the authors do not describe in much detail about how they collected chick data. They mention that chicks are precocial and very small when they join the main group and are 65% adult size at one year (Discussion, sixth paragraph). There is likely a giant difference between having a two-week old chick in the group versus a 6-month-old chick. The authors need to add details on what they consider to be the "chick" stage. Another issue that I see with the chick data is that the authors have controlled for the number of chicks present in groups to see how chicks might affect movement behaviors. Have they tried controlling instead by the presence or absence of chicks rather than the total number of chicks? This may be more logical, as the main group may only be able to move as far or as fast as the member with the shortest legs and having more chicks may not substantially affect those distances/speeds. Finally, do the authors have any information on survival of chicks that could be used as a measure of the success of the groups?

2e) Subsection “Statistical analysis”: You mention that you expect a non-linear relationship because there should be an optimal group size. A linear relationship would mean that increasingly larger group sizes would lead to increasing movement home range sizes. I think it might be worth explicitly mentioning this link between optimal group size and the expectation of a non-linear relationship at the end of your Introduction, where you mention your predictions. It may not be apparent to everyone that an optimal group size would lead to non-linear relationship.

3) Finally, there were questions regarding data or analyses not presented in the paper.

3a) Did you try to include any measure of environmental characteristics in your models, such as resources or temperature? I know that your aim was to examine how group size influences home range size and movement characteristics, but inclusion of information about the habitat of each group would allow you to account for potential environmental differences. Based on Figure 1, it looks like the groups are in relatively close proximity, so this environmental variance may not be large, but I feel this is still worth exploring.

3b) How much variance in home range size, home range overlap and daily distance travelled were explained by the number of adults, chicks etc.? Including this information (e.g., r squared value) would help the reader understand hoe biologically significant these factors are.

---

## [Author Response]

Essential revisions:1) The main revision regards the conclusions drawn in the paper from the data. In part, there are two interacting issues: the framing of the paper and the data support for the optimality conclusion. The theoretical premise of optimal movement does not seem supported in general, and no specific support for the study species is provided. Specifically, the assumption that larger home ranges are better is not well supported and contradicted in some parts of the text. The other predictions (greater variability day-to-day and faster on-the-go movement) could be plausibly argued either way, and no support for the predictions specifically with regard to guinea fowl is presented. Interestingly optimality of fitness (which is mentioned with no support in the text at several points) could actually be assessed using proxies of fitness, such as number of chicks, which are available in the dataset but not used for this purpose in the analyses.

We are grateful to the reviewers that outlined this weakness of the previous version of our manuscript. We have now included one extra analysis showing that groups of intermediate size also have more chicks (Figure 2D). The size of our study groups and the number of chicks per group were already presented in the Supplementary file 1A and in Figure 2 (now Figure 3) of the previous version of our manuscript. Thus we only had to examine this relationship using the same statistical approach as for our other tests, and visualize it with a plot.

We now also provide more information on the biology of our study species (see for example the sixth paragraph of the Introduction, subsection “Study site and study species”, and the fifth paragraph of the Discussion) and how our emerging understanding of our study species drove us to develop an a priori set of predictions: that the costs and benefits of various group sizes, with respect to movement, should result in a non-linear relationship between group size and movement characteristics that reflect an optimal group size. We made several changes along the text, first by re-focusing more on the non-linear relationship between group size and collective movement, second by better justifying the optimality conclusion, and third to avoid claiming that we found the optimal group size for movement but rather that we found evidence that an optimal group size might exist. For a quick look on that see the first and the fifth paragraphs of the Discussion.

Please see individual comments for potential resolutions below, but either a restating of the conclusions or a re-analysis of existing data to support conclusions is needed.1a) The blanket statement that larger home ranges are better is not justified. Large territories or ranging can be signs that things are going wrong (ranging increased for scarce mates, or because resource shortage). This is acknowledged explicitly in the text about the study species, indicating that birds will travel far distances to gain access to the river in time of drought (subsection “Study seasons”). The evidence provided for larger home ranges being better mentions that Florida scrub jay groups that produce more offspring have larger territories (Introduction, third paragraph). Do they maintain larger territories because they are better groups? This question is actually partially answered in the fifth paragraph of the Introduction. The end result is all these statements undermine the key prediction and conclusions of the paper.

We thank the reviewers for focusing on this aspect of our study that seemed poorly-framed and contradictory. These critiques led us to provide greater detail on why we first thought that a larger home range size should be beneficial in our study species. Even though in our Introduction we had a paragraph discussing explicitly the benefits of large home range size (now Introduction, third paragraph) and in our Discussion we discuss again these benefits (see for example Discussion, second paragraph), the reviewers still felt that we seem to contradict our points in some parts of the text and we tried to clarify these parts. We also added an additional aspect in this discussion that we had neglected in our previous version, and has to do with the social benefits of a large home-range size of our study species, which lives in a multilevel society. Potentially this point is important for many other species that live in a similarly organized society (see for example the fifth paragraph of the Discussion).

In summary, we had indicated that larger home-range size can be better for two reasons: a) access to a greater abundance and/or diversity of food resources and b) being less predictable to predators. However, as the reviewers mention, a large home range size can be an indication of animals using an area poor in resources. Our counter-argument on this is that our study guineafowl population lives in area of ca 500ha (=5sqkm) in which home ranges largely overlap. Vulturine guineafowl also express any signs of territoriality, and many groups forage together on glades (open areas rich in resources; see subsection “Study site and study species”). Thus, we have no reason to think that ranging behaviour should be restricted by resource shortage in our study. This is why we argued that our system is ideal for studying the link between group size and movement ability in these landscapes. We also note in the manuscript (subsection “Study seasons”) that in the current study we only considered seasons of intermediate conditions, during which groups do not have to travel to water bodies such as the river.

We also now include an additional argument in our manuscript. This has to do with the social aspect of having a large home-range. VGF live in a multilevel society of stable groups that overlap in space and time with other groups and associate with them preferentially. Larger home range size implies that a group has more chances to associate with other groups on glades and communal roosts. We have not yet quantified the exact benefits of inter-group associations but we do know that sub-adult individuals from one group may disperse to a neighbouring one (dispersal opportunities), they share information on resources and they even breed with individuals from neighbouring groups (mating opportunities) (Introduction, third paragraph)(Grueter et al., 2020).

1b) Bottom line is that the question "Optimal for what?" should be answered. There is clearly an intermediate pattern, but that doesn't necessarily indicate optimality. If we think of optimal foraging theory for instance, there are clearly curved distributions say, in handling time or movement, but what makes them optimal, say for energy, is that there is a measure (usually calories) that coincides with that intermediate peak. That is what is being optimized (energy) is clear, can be objectively measured (calories), and has a clear effect on body condition or fitness. In this case, it is not clear what the movement is being optimized for. A potential way to resolve this issue is to provide some evidence that the factors affecting home range size mentioned in the Introduction actually affect guinea fowl. Is predation indeed higher in birds with more predictable routes? Are larger home range sizes measurably better in terms of resources, bird condition, offspring production, etc? Ultimately, if the desire is to connect movement to fitness (the ultimate currency, in evolutionary terms) then you could measure number of chicks as a measure and these data seem to be available, which is indeed rare data to have! The more I think of it, the more it seems like a lost opportunity, especially when there are many mentions of fitness and movement relating to fitness, but no data to support the linkage (e.g. Discussion, seventh paragraph). Otherwise, the conclusions should be limited to how the size/composition of groups shapes space use.

We again thank the reviewers for this important point. We have now added an analysis showing that groups of intermediate size – matching the peak for movement – have more chicks, which is a good proxy for fitness.

The “Optimality for what?” question is now clearly answered in our Introduction and in the Discussion, for examples: “As a result of the interaction between these two effects, we expect that there will be a non-linear relationship between group size and movement, which, if it results in more effective use of available resources, could correspond to an optimal group size for group movement characteristics. ” And: “Finally, we predict that a non-linear relationship between group size and movement should translate to corresponding peak in fitness, measured as the number of chicks in a group. Such a pattern would suggest that maximising the difference between navigation and coordination represents an optimal group size at which groups are most efficient in exploiting their environment.”

1c) Optimal group sizes: If intermediate sizes are optimal, why should you expect to observe wild groups to be smaller or larger than the optimal? There's a nice description of why group sizes that are unstable at the end of the paper (Discussion, seventh paragraph) – moving some of this description to the Introduction would help set up the ideas tested in the study more strongly (rather than waiting until the end of the paper). Related to this point: Why is Figure 3 in the Discussion? This should be a major part of the paper given the title and it is rarely appropriate to have figures presenting primary data in the Discussion.

We moved Figure 3 to the Results section together with the additional analysis on the number of chicks as a function of group size. We have also added a small section in the Materials and methods section for this part. Because in revising our manuscript we downplayed the topic of optimal group size in our Introduction, we chose not to move this topic out of the Discussion. Instead, we moved the paragraph, to earlier in the Discussion (seventh paragraph).

1d) Why was speed on the go only presented in supplementary material? Even if they don't support the predictions they should be in the main text. Actually, especially if they don't support the predictions.

The results of the model were not presented in the main text, in Table 1, because no model gave significant results for the speed while travelling analysis. As reviewers reasonably suggest though we now present the model with lowest AIC, despite the results being not significant.

1e) The data on chicks slowing down movement is very important and not addressed enough in the text. Together with the data on chick productivity and group size distributions (which for some odd reason is in the Discussion) there could be really interesting insights on questions that are not the focus of this paper for which the data exists to test.

We didn’t find evidence that chicks slow down groups but that in groups with many chicks, home-range size is restricted. We added more on this in the first paragraph of the Discussion.

2) The reviewers had a number of questions regarding potential confounds in the study or factors to control for. These are very important to verify, although I suspect many of these can be addressed rather quickly. Note that some of these points (e.g. group success) have a bearing on larger issues in the manuscript raised above. In any case, they should all be reported on in the final manuscript.2a) Did the number of individuals tagged in a group have any impact on the results? Was this accounted for in the models?

Overall, 75 to 100% of individuals per group wear colour bands and each group carries 1 to 6 GPS tags. However, in this study we only chose randomly one individual per group to represent its group’s movements and thus the number of individuals tagged per group shouldn’t have an effect on our analyses. It is feasible to assume that one individual can represent its group’s movements as typically group members are within 29m of each other 95% of the time tracked (Papageorgiou et al., 2019) (for more explanation on this please see the first paragraph of the subsection “Movement analysis”).

2b) There may have been different environmental conditions across the different years that might have influenced the group movements. Did you account for the different years or for the 5 different seasons in your models, such as a random effect?

Our five study seasons all had very similar environmental conditions (e.g. precipitation, greenness of the habitat, temperature) as described in the *Study seasons* section. GEEs only allow accounting for repetitive measures but not for random effects. Exactly for this reason we picked seasons very similar to each other and accounted for Group ID on the GEEs.

2c) Figure 1: Are all 21 groups represented in each of the 5 seasons? Looking at Season A, there seem to be only 5 groups. If there are a different number of groups per season, do you pool the data for the analyses? If so, do you account for the different number of samples across the different groups, such as weighting the values?

Not all groups are represented in all seasons but as explained above in the GEEs we account for Group ID.

2d) Treatment of chicks: The authors find that the presence of chicks affects group movements. However, the authors do not describe in much detail about how they collected chick data. They mention that chicks are precocial and very small when they join the main group and are 65% adult size at one year (Discussion, sixth paragraph). There is likely a giant difference between having a two-week old chick in the group versus a 6-month-old chick. The authors need to add details on what they consider to be the "chick" stage.

We added some more information on the text to elaborate more on this, as the reviewers reasonably asked for. In the subsection “Study seasons” we give more information on the breeding seasons: “Breeding seasons can occur twice a year (during April or May and during October or November).” we also added in the Materials and methods information on the chicks’ age, “We investigated the relationship between group size and reproductive success by fitting a GEE containing group size and it’s quadratic term as predictors of the median number of chicks in the group in a given season. […] During the rest of the three study seasons (Seasons B, C and E) the chicks where between three and five months old, and each season represented a different cohort of chicks.”

And regarding counting the number of chicks per group we added a section on census data: “Census data was collected daily from a vehicle driving along the roads in our study area. […] Counts are split into the number of adults and the number of chicks, with chicks being easily distinguishable until ~9 months of age.”

Another issue that I see with the chick data is that the authors have controlled for the number of chicks present in groups to see how chicks might affect movement behaviors. Have they tried controlling instead by the presence or absence of chicks rather than the total number of chicks? This may be more logical, as the main group may only be able to move as far or as fast as the member with the shortest legs and having more chicks may not substantially affect those distances/speeds.

Given that there is high variation in the number of chicks per group (range: 0 to 27) we thought that the more realistic way to treat chicks would be as a continuous variable. Based on our field observations, the number of chicks per group affects ranging behaviour.

Finally, do the authors have any information on survival of chicks that could be used as a measure of the success of the groups?

Unfortunately we don’t have that high-resolution data on chick survival for all the study groups, in large part because mortality occurs in the first weeks of life and also because we are still fine-tuning our ability to trap and mark chicks early in their lives. Thus, we instead used the number of chicks per group, when they are three to five months old, as a function of group size (see responses above).

2e) Subsection “Statistical analysis”: You mention that you expect a non-linear relationship because there should be an optimal group size. A linear relationship would mean that increasingly larger group sizes would lead to increasing movement home range sizes. I think it might be worth explicitly mentioning this link between optimal group size and the expectation of a non-linear relationship at the end of your Introduction, where you mention your predictions. It may not be apparent to everyone that an optimal group size would lead to non-linear relationship.

The reviewers are right. We thus changed a sentence in the Abstract from “If these two facets interact, we should observe an optimal group size for collective movement” to “If these two facets interact, we should observe a non-linear relationship between group size and collective movement.” Also, in the Introduction we added: “Given that both positive and negative relationships between group size and home-range size may exist, we a priori predict a non-linear relationship between the size of stable groups and collective movement characteristics. This non-linear relationship may drive an optimum group size for movement that reflects the point where the difference between navigational accuracy and coordination problems is maximized” And: “As a result of the interaction between these two effects, we expect that there will be a non-linear relationship between group size and movement, which, if it results in more effective use of available resources, could correspond to an optimal group size for group movement characteristics. ” We used “non-linear relationship” in several parts across the main text.

3) Finally, there were questions regarding data or analyses not presented in the paper.3a) Did you try to include any measure of environmental characteristics in your models, such as resources or temperature? I know that your aim was to examine how group size influences home range size and movement characteristics, but inclusion of information about the habitat of each group would allow you to account for potential environmental differences. Based on Figure 1, it looks like the groups are in relatively close proximity, so this environmental variance may not be large, but I feel this is still worth exploring.

To be able to focus only on the effect of group size and composition on collective movement avoiding the confounding effect of seasonality and other environmental conditions we focused on seasons (i.e. two-month spans) with very similar conditions. We also added that “Given that our entire study population inhabits a small area, approximately 500ha, and that all groups can easily access all parts of our study area, we assume that groups’ relative ranging behaviour was not restricted by the environment during these intermediate seasons.”

3b) How much variance in home range size, home range overlap and daily distance travelled were explained by the number of adults, chicks etc.? Including this information (e.g., r squared value) would help the reader understand hoe biologically significant these factors are.

We calculated the R square of each GEE tested based on (Zheng, 2000) and we now present it in all tables at the Supplementary file 1B to F and on Table 1.